# Hamstring Strain Injury Patterns in Spanish Professional Male Football (Soccer): A Systematic Video Analysis of 78 Match Injuries

**DOI:** 10.3390/jfmk10020201

**Published:** 2025-05-31

**Authors:** Aitor Gandarias-Madariaga, Antonio Martínez-Serrano, Pedro E. Alcaraz, Julio Calleja-González, Roberto López del Campo, Ricardo Resta, Asier Zubillaga-Zubiaga

**Affiliations:** 1Department of Physical Education and Sport, Faculty of Education and Sport, University of the Basque Country (EHU), 01007 Vitoria-Gasteiz, Spain; agandarias004@ikasle.ehu.eus (A.G.-M.); julio.calleja.gonzalez@gmail.com (J.C.-G.); 2Centro de Investigación de Alto Rendimiento Deportivo UCAM, Universidad Católica San Antonio, 30830 Murcia, Spain; amartinez30@ucam.edu (A.M.-S.); palcaraz@ucam.edu (P.E.A.); 3Sports Research Area of LaLiga, 28043 Madrid, Spain; rlopez@laliga.es (R.L.d.C.) rresta@laliga.es (R.R.)

**Keywords:** soccer, muscle injury, injury mechanisms, sprint patterns, video analysis

## Abstract

**Background**: To the best of the authors’ knowledge, the different injury mechanisms of the hamstring muscle group have not been defined in detail in men’s professional football. For this reason, the main aim of this study is to determine the mechanisms and contextual patterns associated with hamstring muscle group injuries in professional male football players in competition, using a systematic video analysis method. **Methods**: Video recordings of official matches from two consecutive seasons (2017/18 and 2018/19) of the Spanish First Division of Football (LaLiga^TM^) were used for this research. The process to determine the moment of injury was carried out by two independent evaluators using an ad hoc observation tool and, subsequently, all relevant data were collected to detail the specific patterns of injury events observed. **Results**: In total, 78 cases of hamstring injuries were included for the final analysis of specific patterns. The most outstanding results were that, (1) although the sprint-related pattern (SP) is predominate (54%; 42 cases), the combined pattern 2 (COMB2) is another mechanism that appears frequently (26%; 20 cases), (2) within the SP, curved runs show a greater number of cases (52% of SP; 22 cases), (3) the majority of the injuries occur without contact (83%; 65 cases) and with the presence of the ball (88%; 69 cases), and finally, (4) the most injured positions were fullbacks/wingbacks (28%; 22 cases), central defenders (27%; 21 cases), and wingers/wide midfielders (23%; 18 cases). **Conclusions**: The SP remains the most frequent pattern in hamstring injuries; however, the present study presents other mechanisms that are also quite common and should be considered, such as curvilinear runs in SP and COMB2.

## 1. Introduction

Player availability, defined as keeping players injury-free and ready to participate in competition, is extremely important in the current elite team sport scenarios because it is related to team performance [1] and has been demonstrated to have a close relationship with the number of goals scored [2] and points obtained throughout the season [3], thus determining ranking position [4]. Therefore, injury prevention strategies and their management becomes a cornerstone towards the achievement of these outcomes [5,6]. However, despite of the overall injury rate shows a downward trend, muscle injury rate has not decreased and neither has its injury burden [7]. Hamstring strain injury (HSI) is considered the most frequent muscle injury in professional football, accounting for about 40% of muscle injuries [8,9,10] and between 12% and 19% of the total injuries [8,9,10,11,12] showing an injury incidence approximately 10 times higher in matches than in training (4.99/1000 h vs. 0.52/1000 h, respectively) [11].

The hamstring muscle group is composed of the muscles located in the posterior and medial area of the thigh, which are the semimembranosus (SM), semitendinosus (ST), and the biceps femoris (BF) on the lateral side, both its long (BFlh) and short (BFsh) heads [13,14]. Hamstrings are involved in a large number of athletic movements (e.g., running, jumping, deceleration, landing and shooting), so their function is essential in the physical performance of football, especially when high-speed running is required [15,16,17]. Therefore, it is not surprising that most studies concluded that HSIs occur during this type of action [16,18,19,20,21,22,23,24,25,26,27,28,29].

Regarding the hamstring injury mechanisms, Askling et al. [21,30] proposed two specific types of injury (stretch-type and sprint-type); however, it is not common to find detailed descriptions in the scientific literature about injury events in professional football players. Recently, two studies [28,31] have implemented video analysis to more thoroughly define the HSIs in men’s professional football. The use of this methodology allows for understanding and expanding information about different injury mechanisms (e.g., hamstring, adductor, and anterior cruciate ligament), as well as providing information about how the injury occurs in a real game context [28,31,32,33,34,35,36]. The two studies cited above [28,31] categorized the hamstring injury mechanisms, defining different subtypes within the patterns related to sprinting (acceleration, high-speed running, deceleration, and high-speed curved running) and related to stretching (open chain and closed chain).

However, Jokela et al. [31] presented the importance of including mixed patterns (associating the sprint-pattern with the stretch-pattern) since their neuromuscular characteristics are different. In addition, the high neurocognitive and motor control demands implicit in football as mechanisms that could be harmful and that appear recurrently in football players, external focus of attention, unexpected disturbances, etc., should be considered [37,38,39]. Therefore, the main aim of this study was to describe the mechanisms and contextual patterns related to HSIs of professional male football players in competition, using systematic video analysis.

## 2. Materials and Methods

### 2.1. Study Design

This descriptive observational cross-sectional study included all male football players of the Spanish First Division (LaLiga^TM^) of the 2017/18 and 2018/19 seasons who played at least one official competition match as inclusion criteria. Personal data (name, date of birth, club, and playing position) of the participants were provided by LaLiga^TM^ (LFP) through the Mediacoach^®^ application (official data provider of LFP, from open access websites such as https://www.transfermarkt.es (accessed on 20 June 2021) (Transfermarkt, Hamburg, Germany), and if necessary, by performing an additional Google^®^ search following a methodology similar to the one used by Gronwald et al. [28] and Klein et al. [35].

#### Ethical Considerations

All data used in this study were treated confidentially, and players’ personal information was obtained through publicly available websites, such as https://www.transfermarkt.es (accessed on 20 June 2021) (Transfermarkt, Hamburg, Germany), so ethical clearance was not required [33,40].

### 2.2. Procedures

#### 2.2.1. Video Acquisition and Processing

Before starting to analyze the observability of injuries, with data obtained mainly through https://www.transfermarkt.es (accessed on 20 June 2021) (Transfermarkt, Hamburg, Germany), all muscle injuries from the 2017/18 and 2018/19 seasons of LFP were entered into a Microsoft^®^ Excel table. Once all injuries were added, those that met the following criteria were included for analysis: HSIs, that happened in official LFP matches (i.e., excluding training, Copa del Rey, Champions League, etc.) or that some information about the injury could be found. Subsequently, a more thorough screening was carried out, looking at each recording to see if the moment of injury was observable or not through the Mediacoach^®^ platform. After this screening, the most exhaustive analysis was performed with the remaining injuries, cutting complete fragments from the previous play until the injury occurred, reviewing as many times as necessary (different angles and slow motion) from the different cameras available (television and tactical camera).

#### 2.2.2. Determination of the Moment of Injury

The moment of injury was assumed to occur when the player clearly showed some gesture of discomfort or inability to continue in the game (e.g., if the player started to limp or put his hand on the back of his thigh immediately after the observed action). This process was carried out by two operators (A.G.-M. and A.M.-S.) and was performed by testing with smaller samples (first with 25 cases and then with 40) to ensure that the two observers agreed on the time of injury. If the observers did not agree with any of the cases, the injury was rechecked to reach an agreement with the help of a third author (A.Z.-Z.). All those cases in which no consensus was reached were excluded from the final analysis.

### 2.3. Video Analysis

To perform the analysis of the videos, an ad hoc tool was first created for the observation of HSIs in football players. This tool has a system of field formats that is structured in eight criteria, which, in turn, are composed of several categories following the principles of exhaustiveness and mutual exclusivity (EME) [41]. To develop the different criteria and categories of this tool, previous studies with similar objectives were used as a reference [28,35]. To assess the reliability of the tool, inter- and intra-observer reliability analyses were performed independently [42,43] using IBM SPSS^®^ V26 and obtaining high levels of reproducibility: Cohen’s Kappa >0.87 and *p* < 0.05 in all cases. For more information about the different criteria and categories used for the analysis of the videos consult Appendix A.

Once the observation tool was defined, the evaluation and data recording of the specific injury patterns of the hamstring muscle group were carried out retrospectively by the main author of the study (A.G.-M.). With each injury, data were collected on contextual factors (equipment, specific position, moment…) and injury characteristics (contact, pattern, trajectory, technical action…). Periodically (each 2 weeks), a meeting was held with two of the co-authors (A.G.-M. and A.Z.-Z.) to discuss the collected data and reach an agreement. All these data were annotated in a Microsoft Excel^®^ sheet and, subsequently, descriptive statistics were performed in it.

## 3. Results

Between the 2017/18 and 2018/19 seasons, a total of 733 muscle injuries were reported. After considering the first inclusion criteria, the number was reduced to 125 hamstring injuries. Among these cases, the number of videos for the final analysis was 78 observable injuries (Figure 1).

### 3.1. Subjects

A total of 78 hamstring injuries from 62 professional football players (mean age: 28 ± 3.91 years) were included.

### 3.2. General Descriptive Data of the Injuries

#### 3.2.1. Contextualization

Contact. Eighty-three percent of the injuries (65 cases) were non-contact injuries, with the remaining injuries being contact injuries (17%; 13 cases).

Specific position. The positions that suffered the most from HSIs were especially three, fullbacks/wingbacks (28%; 22 cases), central defenders (27%; 21 cases), and wide midfielders or wingers (23%; 18 cases). The next most numerous cases were central midfielders (10%; 8 cases) and strikers (9%; 7 cases). Finally, there were only two isolated cases of goalkeepers (1%; 1 case) and second strikers (1%; 1 case).

Ball. In 69 cases (88%), the presence of the ball was relevant at the time of injury, while in only 9 cases (12%) it was not.

Time of injury. The time at which injuries occurred during the match was evenly distributed, although it could be said that the block of the middle of the first half (between minutes 16–30, 23%; 18 cases) and second half (between minutes 61–75, 21%; 16 cases) are slightly above the rest. This is followed by the beginning of the second part (18%; 14 cases) and the end of the first part (17%; 13 cases), with smaller cases of the end of the second part (12%; 9 cases) and the beginning of the first part (10%; 8 cases).

Situation. Injuries were distributed almost equally between defensive and offensive actions, the former representing 51% of the cases (40 cases) and the latter 49% (38 cases) (Table 1).

#### 3.2.2. Injury Mechanism

Injury pattern. The most prominent injury patterns were the SP-type pattern, presenting more than half of all cases (54%; 42 cases) and COMB2 (26%; 20 cases). The remaining cases were the closed chain stretch-related pattern (ST-CC) (8%; 6 cases), open chain stretch-related pattern (ST-OC) (6%, 5 cases), and the combined pattern 1 (COMB1) (6%, 5 cases).

Trajectory. Among the SP-type injury, 52% of the cases showed a curvilinear trajectory (22 cases), while 48% showed a rectilinear trajectory (20 cases).

Technical action. As for the specific technical action at the time of injury, the cases were mainly divided into three groups, the action “none/stopped” (37%; 29 cases), the “dispute” (36%; 28 cases), and the “pass” (13%; 10 cases). In a smaller percentage were found the action “control” (5%; 4 cases), the “steal” (3%; 2 cases), the “clearance” (3%; 2 cases), the “center” (3%; 2 cases), and the “drive” (1%; 1 case) (Table 2).

### 3.3. Relevant Descriptive Data Within Each Injury Pattern

SP type injury pattern (*n* = 42). Within this injury pattern, the most important data showed that 95% of the cases (40 cases) occurred without any type of contact, mainly in central defenders (33%; 14 cases) and fullbacks/wingbacks (26%; 11 cases), with a significant presence of the ball (79%; 33 cases), at the beginning (minutes 46–60) of the second half (29%; 12 cases), in mostly defensive situations (60%; 25 cases), and without any specific technical action (i.e., running) (60%; 25 cases) and in disputes (38%; 16 cases).

ST-OC type injury pattern (*n* = 5). The most relevant was that all of them occurred without contact and with the presence of the ball, in the middle part (minutes 61–75) of the second half (60%; 3 cases), mostly in offensive situations (80%; 4 cases), and in technical passing actions (60%; 3 cases). Regarding the specific position, 4 of the cases were players with a more defensive role (80%; 4 cases, 2 central defenders, 1 goalkeeper, and 1 fullback).

ST-CC type injury pattern (*n* = 6). Again, 100% of the cases occurred with the presence of the ball, mainly during the middle part (minutes 16–30) of the first half (50%; 3 cases), in mainly offensive situations (67%; 4 cases), and in contested actions (67%; 4 cases). As for contact, the cases were equally distributed (50% without contact and 50% with contact). The position of the players, in this case, was more offensive (100%; 6 cases, 4 wide midfielders/wingers, 1 center striker, and 1 central midfielder).

COMB1 type injury pattern (*n* = 5). Once again, the presence of the ball was relevant in all five cases, almost all were non-contact injuries and in defensive situations (80%; 4 cases), and throughout the middle part of the first half (60%; 3 cases). The role of these players was of a more defensive nature (80%; 4 cases, 2 central defenders and 2 fullbacks), and the specific action was distributed (2 cases of “control”, 1 of “dispute”, 1 of “none”, and 1 of “pass”).

COMB2 type injury pattern (*n* = 20). In this type of injury, the presence of the ball again prevailed (100% of cases), the vast majority occurred without contact (65%; 13 cases), mainly during offensive situations (60%; 12 cases), in fullbacks (40%; 8 cases), and in contested and passing actions (35%; 7 cases and 30%; 6 cases, respectively). The time of injury was distributed between the first and second half, although minutes 16–45 prevailed (50%; 10 cases). Although some trends could be observed, it should be noted that this is possibly the pattern in which there was the greatest dispersion of data (Table 3).

## 4. Discussion

The purpose of this study was to describe in detail how hamstring strain injuries occurred in male football players of the Spanish football league, during official matches in two consecutive seasons. The most significant findings were that (1) although the SP still stands out over the others, COMB2 is also another mechanism to be taken into account, (2) within the SP-type patterns, curvilinear running can be an interesting stimulus to consider in training programs, (3) most injuries occur without contact and with the presence of the ball, and (4) the specific positions most affected are fullbacks/wingbacks, central defenders, and wide midfielders/wingers.

### 4.1. Sprint Pattern (SP)

The scientific literature has described the hamstring injury mechanism primarily through two mechanisms of injury, the stretch type and the sprint type [21,30]. Many researchers agree that the sprinting or high-speed running type is, between the two, the one that occurs more frequently and generates more cases of hamstring injury [16,18,19,20,21,22,23,24,25,26,27,28,29]. Based on several biomechanical studies analyzing the demands on the hamstring musculature during high-speed running [44,45], despite some discrepancies between the mechanics of the acceleration phase and maximum speed, it is considered that the most critical moments for injury are the final phase of the swing (preactivation) and the early stance phase (braking), in the first case braking the inertia produced by the lower limb and in the second the ground reaction forces [16,19,22,23,29,46,47,48,49,50].

Our results show that more than half of the cases (54%) occurred during pure running actions at high intensity, and, moreover, 86% of all cases (42 pure SP cases, 20 COMB2 cases, and 5 COMB1 cases) were related in some way to sprinting, as COMB1 and COMB2 patterns involve a sprinting action prior to the time of injury. Other relatively recent studies also using systematic video analysis to describe hamstring injury in soccer and rugby players show similar findings, as in all of them sprinting is the predominant mechanism of injury. Gronwald et al. [28] showed that 25 of 52 cases (48%) occurred during sprinting, Kerin et al. [32] noted that 8 of 17 cases (47%) were during running, Jokela et al. [31] noted that 3 of 14 cases (21%) were pure sprinting, although similar to our study, six cases (43%) also had a relationship with sprinting, as they were “mixed type” patterns (so in total 64% of cases were related to sprinting). Finally, the study by Aiello et al. [51] showed that 16 of 17 cases (94%) of hamstring injury occurred while the players were running at a speed above 25 km/h, all of them at speeds above 70% of the players’ individual maximum speed. Therefore, considering these previously commented data, sprints or high-speed running should continue to be an important pillar within the rehabilitation and prevention programs for hamstring injury, since it is the action with the highest incidence of this injury.

### 4.2. Combined Pattern 2 (COMB2)

The findings of this study show that the second most recurrent injury pattern was COMB2, presenting 26% of all injuries observed. It is relevant to emphasize the incidence of COMB2 in hamstring injury, because although it involves a sprinting action, the mechanics and neuromuscular demands are different since the injury might not occur in the sprint as such, but when the lower extremity has to decelerate rapidly the inertia of the body, accompanied in some cases by external forces generated by an opponent. Along these lines, some recent studies in men’s football and rugby, which also used systematic video analysis [28,31,32,51], have found that braking can be an important mechanism of injury. Moreover, similarly, although in their investigation the total number of cases was smaller (*n* = 14), Jokela et al. [31] found six injuries that they term the “mixed type” pattern, of which four (29%) were of the COMB2 style. For this reason, these authors also highlight the relevance of recognizing and incorporating biomechanical studies of this “mixed type” injury pattern, as its prevalence is high and does not follow the typical patterns of sprint or stretch type injuries. Of the other three studies, two [28,32] had a different way of categorizing injuries, mainly because they do not consider combined patterns, so the results obtained in their studies differ slightly. Even so, Gronwald et al. [28] identified that 18 cases (%35) of 52 hamstring injuries occurred in landing and stride type braking, and Kerin et al. [32] that 3 (18%) of 17 cases were generated in sudden decelerations. Finally, Aiello et al. [51] observed that players frequently (4 out of 17 cases; 24%) injured their hamstrings when they were decelerating from high speeds, more specifically when running on average at an intensity of 88% of their individual maximum speed. Considering these data, it could be concluded that the physical ability to be able to break the body quickly (and often with some disturbance generated by the opponent) should be a key variable to take into account in training programs for the prevention of hamstring injuries.

### 4.3. Curvilinear Sprints

Within pure sprinting patterns, the importance of curvilinear sprints should be noted, since in our study we observed that approximately half (52% of the cases observed) of the injuries occurred in curvilinear sprinting. It has been described that the ability to sprint in curvilinear trajectories is an important skill in football [52,53,54,55] which is used to evade, chase, or lure an opponent. Likewise, it has been observed that football players who run fast in linear sprints do not necessarily run faster in curvilinear trajectories [53,54,55]. This may be because curved sprints require the ability to generate centripetal forces, causing different mechanical and neuromuscular behaviors [53]. For example, several studies have been able to identify that the outer and inner legs do not act in the same way, increasing the contact time and decreasing the step length and step frequency of the inner leg [53,54,55]. Moreover, in terms of electromyographic activation, it appears that in the outer leg (performing a constant side-stepping maneuver) the hip external rotators (biceps femoris and gluteus medius) are more active, while in the inner leg (performing a constant cross-stepping maneuver) are the hip internal rotators (semitendinosus and adductors) [55]. These could be some of the reasons why there is a “good” and a “bad” side to curved sprints.

However, there seems to be controversy among the different observational studies analyzing the sprint trajectory. In the study by Jokela et al. [31], although the total number of pure sprint injuries was low, they saw that 1 of 3 injuries (33%) was in a curvilinear sprint. Nevertheless, Aiello et al. [51] indicate that 88% of the injuries analyzed were in rectilinear sprints. Other research points out that approximately 85% of the maximum sprints in professional football present some degree of curvature, i.e., they are not totally linear [53,55]. These differences between studies may be due to the fact that the methodology used for the analysis of the trajectory of the sprints was different, details such as the total number of cameras used, their angulations, or references used to measure the degree of curvature, could be some of the reasons for these differences. Despite these differences, physical trainers and coaches of football players should implement curvilinear sprints in their training and prevention programs, in order to work on the distinctive neuromuscular mechanics and demands that they require.

### 4.4. Presence of Contact and Ball

Some epidemiological studies conducted in men’s football report that the vast majority of muscle injuries, and specifically hamstring injuries, occur in non-contact situations [8,56]. In both this present study and two other studies using video analysis system in team sports [28,32], a large proportion of the injuries analyzed are still non-contact (83%, 65%, and 59%, respectively), although in the research by Jokela et al. [31], the number of cases decreases to 50%. This could point out that as the sample of hamstring injuries increases, so does the number of cases of non-contact injuries. However, both in our case and in the study by Gronwald et al. [28], contact injuries increase in patterns involving stretching, mainly closed chain, 59% (10 of 17 cases) and 72% (13 of 18 cases) of cases involving contact, respectively. Likewise, based on our results, in both ST-CC and COMB2 pattern type injuries, the technical action of “dispute” predominates. Relating the last two points, in football there are situations in which it is necessary to collide with an opponent in order to get the ball, and as has been previously highlighted, this can generate large external forces that must be stopped, especially if the two players are running at high speed.

In our study we have observed that the only pattern that in some cases does not show the presence of the ball is the sprint pattern, while all the other patterns are totally influenced by it. The presence of the ball determines whether or not the ball was involved at the time of injury, in other words, whether its presence was relevant in affecting the mechanism of injury. Disputing against an opponent to get possession of the ball or trying to control a ball that comes from a long pass from a teammate, could be examples in which the ball would have a great influence on the mechanism. Likewise, the technical actions that have the highest relationship with this variable are “dispute” (36%; 28 cases), “none/stopped” (26%; 20 cases), and “pass” (13%; 10 cases). Other studies also using systematic video analysis to investigate different types of injuries occurring in football [28,31,33,35] do not consider the presence of the ball as a factor that may influence the mechanism of injury, so we consider it as original and distinctive to this study. The only study that considers the ball as an analysis criterion is that of Aiello et al. [51]; however, they observed whether or not the injured player was carrying the ball at the time of the injury. It is important to clarify that our study did not analyze this criterion in this way, since in our case the player did not necessarily have to carry the ball, in other words we observed whether the activity close to the ball could influence the injured player’s action.

This could be related to the dynamic nature of team sports such as football, in which the environment is constantly changing, involving high neurocognitive and motor control demands [37,38,39]. Consequently, football players must attend to complex visual demands involving an external focus of attention (e.g., looking at the ball), unanticipated perturbations (e.g., an opponent’s push), and other interactions with the environment, and may affect the complex integration of the sensory system (vestibular, visual, and somatosensory), ultimately altering the response of the neuromuscular system [37,38,39]. Although these studies focus on anterior cruciate ligament injury, perhaps it might also be interesting to include this approach during the later stages of hamstring injury prevention programs, where tasks contain greater neurocognitive and motor control demands.

### 4.5. Specific Position

There are several studies that have analyzed the different physical demands differentiating by specific playing position of professional male football players in competitive games [57,58,59,60,61,62]. Almost all agree that midfielders, in general, are the players who travel the greatest total distance [57,58,60,62]. However, in terms of high metabolic demand actions (measured by metabolic power and high metabolic load distance), i.e., actions involving a high density of high-speed running and accelerations/decelerations, although there are some discrepancies between studies, it seems that wing players (wingers and fullbacks/wingbacks) and center midfielders are the ones that prevail over the others [57,58,59,60,61,62]. In contrast, most scientific papers also agree that the players with the lowest metabolic demand are central defenders and forwards [57,58,60,61], although it should be noted that some of them also show that forwards perform a large number of accelerations and sprints, but with more stop-start type runs [57,62]. However, it should be considered that such physical demands are directly conditioned by the style of play of the teams, which could explain the differences between studies investigating these variables [57,62].

Contrasting these results with our study, we can observe that wing players such as fullbacks/wingbacks and wingers are one of the positions most affected by hamstring injury, something that is not surprising considering the data previously discussed [57,58,59,60,61]. However, the third group that prevails over the others is that of central defenders, an interesting fact considering that it is one of the positions with the lowest physical demands in men’s professional football [57,58,60,61]. Although probably in a more isolated way, these players also have to perform high-intensity actions, for example, to dispute ball possession against more offensive players (wingers and forwards), generating situations with high intensity running requirements [57]. One of the possible reasons for the high incidence in these players could be related to the differences in physical load demands between training and matches. In this direction, several studies show that there are evident differences between training and matches in terms of physical demands, indicating that, usually, during training, the needs of high-speed running and sprinting are not sufficiently trained, something that may be associated with the preferential use of small-sided games tasks [62,63].

### 4.6. Strengths and Limitations

This study provides a large sample of hamstring injuries sustained during official men’s professional football matches. Systematic video analysis has been carried out using a standardized process (observability of injuries, observation tool, defining the time of injury…) involving several experts from the field of sports science, a methodology that has already been used in several previous studies examining different types of injury [28,31,32,33,34,35,36,40].

However, there are some limitations. First, the main limitation of this study is that it is purely descriptive in nature. For this reason, the next step and line of research could involve conducting a more in-depth analysis of the data, potentially interrelating the findings of this study with other variables associated with injury risk (such as age, playing time, field position, etc.). In this regard, the primary constraint has been the lack of available data—for example, from other devices (GPS/GNSS, gyroscopes, etc.) commonly used today to quantify player load, or access to official medical reports from soccer clubs. Such data would have allowed for a more comprehensive injury analysis.

Second, although the final sample of injuries is considerable, many injuries (47 cases out of 125; 38%) had to be discarded because of unobservability and because we could not establish a consensus on the mechanism or timing of the injury. Since this is a visual analysis, the number of cameras available, their recording quality, and angulation are decisive. In addition, visual analysis of the injury makes it difficult to determine the specific event that triggered it. Nevertheless, it is possible to establish criteria based on the player’s reaction (gesture of discomfort, limping, hand on thigh…).

Third, this study provides new insights into previously undescribed mechanisms of hamstring injury in men’s professional football. It would be interesting (and of great relevance) to analyze if these patterns are reproduced in other populations: other categories, female players, other team sports.

## 5. Conclusions

In summary, this observation study reveals that while the SP remains predominant, the COMB2 pattern warrants significant consideration due to its unique physical demands. Within the SP, the importance of curved sprints is highlighted, as their neuromuscular demands differ markedly from straight sprints.

As with most muscle injuries, a large proportion of the injuries analyzed in this study were non-contact. However, in closed-chain patterns (COMB2 and ST-CC), the number of contact cases increased. Furthermore, the presence of the ball should be a key factor to consider, as it could alter players’ mechanics. In addition, wing players (fullbacks/wingbacks and wingers) generally suffered the most hamstring injuries, though the central defender’s group also merits attention.

Therefore, while hamstring injury prevention programs should continue to incorporate high-intensity linear sprints, may be crucial to also include sprints with curvilinear trajectories and mechanisms that work the COMB2 pattern, given their relatively high observed incidence. Additionally, considering the high neurocognitive and motor control demands placed on soccer players during games, training, and rehabilitation could integrate tasks involving complex environmental interactions. This includes unexpected disturbances or external attention, which provide a challenging component for the sensory system.

## Figures and Tables

**Figure 1 jfmk-10-00201-f001:**
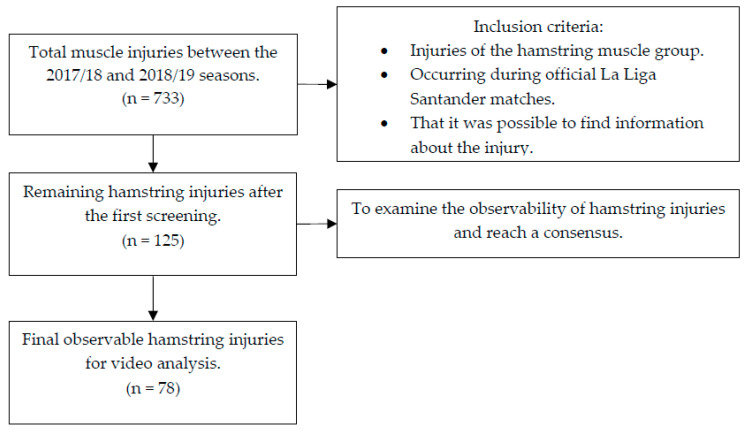
Flow chart describing the acquisition and screening process.

**Table 1 jfmk-10-00201-t001:** General descriptive data of the contextual factors of the injuries.

Contact	No. of Cases	Percentage	Specific Position	No. of Cases	Percentage	Ball	No. of Cases	Percentage	Time of injury	No. of Cases	Percentage	Situation	No. of Cases	Percentage
CI	13	17%	CD	21	27%	NO	9	12%	1 HALF-END	13	17%	DEFEN	40	51%
NCI	65	83%	FB/WB	22	28%	YES	69	88%	1 HALF-MID	18	23%	OFFEN	38	49%
			STR	7	9%				1 HALF-BEG	8	10%			
			WIN/WMF	18	23%				2 HALF-END	9	12%			
			STR	8	10%				2 HALF-MID	16	21%			
			SE-STR	1	1%				2 HALF-BEG	14	18%			
			GK	1	1%									

CI, contact injury; NCI, non-contact injury; CD, central defender; FB/WB, fullback/wingback; STR, striker; WIN/WMF, winger/wide midfielder; SE-STR, second striker; GK, goalkeeper; DEFEN, defensive actions; OFFEN, offensive actions.

**Table 2 jfmk-10-00201-t002:** General descriptive data on the mechanism of injury.

Injury Pattern	No. of Cases	Percentage	Trajectory	No. of Cases	Percentage	Technical Action	No. of Cases	Percentage
COMB1	5	6%	CURV	22	52%	CRO	2	3%
COMB2	20	26%	LIN	20	48%	CON	4	5%
SP	42	54%				DRI	1	1%
ST-OC	5	6%				CLE	2	3%
ST-CC	6	8%				DIS	28	36%
						Ø	29	37%
						PAS	10	13%
						STE	2	3%

COMB1, combined injury pattern 1; COMB2, combined injury pattern 2; SP, sprint-type injury pattern; ST-OC, open-chain stretch injury pattern; ST-CC, closed-chain stretch injury pattern; CURV, curved sprint; LIN, linear sprint; CRO, cross; CON, control; DRI, driving; CLE, clearance; DIS, dispute; Ø, none/stopped; PAS, pass; STE, steal.

**Table 3 jfmk-10-00201-t003:** Notable descriptive data of the different injury patterns.

Injury Pattern		Contact				Specific Position																	Ball			
		CI		NCI		CD			FB/WB		STR		WIN/WMF			CMF			GK		SE-STR		YES		NO	
COMB1 (*n* = 5)		1 (20%)		4 (80%)		2 (40%)			2 (40%)		-		1 (20%)			-			-		-		5 (100%)		-	
COMB2 (*n* = 20)		7 (35%)		13 (65%)		3 (15%)			8 (40%)		3 (15%)		4 (20%)			2 (10%)			-		-		20 (100%)		-	
SP (*n* = 42)		2 (5%)		40 (95%)		14 (33%)			11 (26%)		3 (7%)		8 (19%)			5 (12%)			-		1 (2%)		33 (79%)		9 (21%)	
ST-OC (*n* = 5)		-		5 (100%)		2 (40%)			1 (20%)		-		1 (20%)			-			1 (20%)		-		5 (100%)		-	
ST-CC (*n* = 6)		3 (50%)		3 (50%)		-			-		1 (17%)		4 (67%)			1 (17%)			-		-		6 (100%)		-	
**Moment**										**Situation**				**Technical Action**												
**1 HALF END**	**1 HALF MID**		**1 HALF BEG**		**2 HALF END**		**2 HALF MID**	**2 HALF BEG**		**DEFEN**		**OFFEN**		**CRO**	**CON**		**DRI**	**CLE**		**DIS**		**Ø**		**PAS**		**STE**
1 (20%)	3 (60%)		-		1 (20%)		-	-		4 (80%)		1 (20%)		-	2 (40%)		-	-		1 (20%)		1 (20%)		1 (20%)		-
4 (20%)	6 (30%)		1 (5%)		2 (10%)		5 (25%)	2 (10%)		8 (40%)		12 (60%)		1 (5%)	1 (5%)		-	1 (5%)		7 (35%)		2 (10%)		6 (30%)		2 (10%)
7 (17%)	6 (14%)		5 (12%)		5 (12%)		7 (17%)	12 (29%)		25 (60%)		17 (40%)		-	-		1 (2%)	-		16 (38%)		25 (60%)		-		-
-	-		1 (20%)		1 (20%)		3 (60%)	-		1 (20%)		4 (80%)		-	1 (20%)		-	1 (20%)		-		-		3 (60%)		-
1 (17%)	3 (50%)		1 (17%)		-		1 (17%)	-		2 (33%)		4 (67%)		1 (17%)	-		-	-		4 (67%)		1 (17%)		-		-

COMB1, combined injury pattern 1; COMB2, combined injury pattern 2; SP, sprint-type injury pattern; ST-OC, open-chain stretch injury pattern; ST-CC, closed-chain stretch injury pattern; CI, contact injury; NCI, non-contact injury; CD, central defender; FB/WB, fullback/wingback; STR, striker; WIN/WMF, winger/wide midfielder; CMF, central midfielder; SE-STR, second striker; GK, goalkeeper; DEFEN, defensive actions; OFFEN, offensive actions; CRO, cross; CON, control; DRI, driving; CLE, clearance; DIS, dispute; Ø, none/stopped; PAS, pass; STE, steal.

## Data Availability

The data presented in this study are openly available in Transfermarkt https://www.transfermarkt.es (accessed on 20 June 2021) [Transfermarkt] [https://www.transfermarkt.es (accessed on 20 June 2021)].

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
