# Peer review of "Hamstring Strain Injury Patterns in Spanish Professional Male Football (Soccer): A Systematic Video Analysis of 78 Match Injuries"

_jfmk, 2025, doi:10.3390/jfmk10020201_

Round 1
Reviewer 1 Report
Comments and Suggestions for Authors
Authors have made a very large data collection of videos about hamstring strain injuries (HSI)
during two consecutive seasons within the major Spanish league, which is one of the most
important leagues worldwide, with the aim of reporting the mechanisms and contextual patterns
during the occurrence of HSI. This is a very important topic as HSI are the most common and
burdensome injury type in football, and, as reported by the Authors, the unavailability of players
from HSI reduces the overall competitiveness of a team.
A systematic video analysis of HSI specifically in professional soccer players, to the best of my
knowledge, has been previously done in just three investigations. However, they collected fewer
HSI episodes; thus, a more recent update with a higher number of HSI occurrences was very
warranted. Moreover, the paper under review [jfmk-3592205] went deeper into the mechanisms
and contextual patterns of HSI. Indeed, for example, it differentiates between the sprint type (e.g.
curvilinear vs rectilinear), player position (e.g. fullbacks/wingbacks, central defenders, etc.), the
time of occurrence (e.g. 0-15 minutes, 15-30 minutes, etc.), as well as the presence of the ball. All
these descriptions are very helpful for the take-home message. Thus, in my opinion, I consider this
paper original and relevant to practitioners and researchers.
The key takeaway from this research is directed at the staff responsible for the rehabilitation and
conditioning of football players. It provides clear guidance on which movements should be
incorporated into conditioning programs to enhance hamstring injury resilience, as well as which
player positions may benefit most from specific off-pitch resistance training.
From a methodological point of view, this study applied an identical procedure to other systematic
video analyses of sport injuries reported in the literature; thus, in my opinion, this section could be
judged optimal.
Tables and Figures are clear, well-written and described. References are coherent with the
manuscript and relevant to the topic.
The discussion and the conclusion of the study are consistent with the evidence presented, as they
describe and comment on the type of HSI and compare it with the relevant literature. However, I
noticed that I could have missed some further contextualisation, which is presented below:
Comments:
In the discussion section, can you deepen the goalkeeper HSI? How did this injury happen? Could
be a new study perspective?
Results find that in the middle of the first and the second half, there is a slight increase in HSI. Did
it coincide with the part of the game in which players repeat more high-intensity running and,
therefore, could be more fatigued? Moreover, results find that the second period with more HSI is
at the beginning of the second half. This could be because, on the contrary, players had a lower
muscle temperature (if they did not have a warm-up practice during the recovery time) and,
therefore, their muscles are more susceptible to injuries. Again, the fewer episodes at the end of
the second half could be because there are fewer overall high-intensity actions by the players?
Can you expand on these concepts in the discussion?
Regarding HSI and curvilinear sprint, why was it not reported whether the injury occurred in the
inner or outer leg? Could it be an additional and useful information for future research looking at
which leg is more susceptible to HSI in curvilinear sprint?
What was the age of the injured players? Could it be an useful information to stratify your results?
Do older players suffer prevalently from a certain type of HSI than younger players?
Additionally, some of these HSI were they a re-injury? Again, could it be another variable to stratify
the results?
Finally, considering the easy-to-access data (video recording in open access) and no requirement
of an ethics form, why did you only consider two seasons and not a longer period (such as 5 years)
as in one of the studies cited as reference?
Author Response
First of all, thank you very much for your comments. They provide very interesting additional information for our study, and we will take them all into consideration. We have observed that the vast majority of them refer to the methodological section of the study, so we believe it is appropriate to write this brief introduction to clarify most of the questions raised.
When we began working on this study, the main objective was to describe in more detail the injury mechanisms of the hamstring muscles in soccer players, as until then there was little descriptive information about them. To this end, we contacted LaLiga so they could provide us with videos of some sports seasons through the Mediacoach platform (which includes the tactical and television cameras). They agreed to work with us, but we had a time limit (6 months) to complete the entire video filtering process, include all injuries that were solely related to the hamstring muscles, generate the ad hoc observational tool, analyze the observability of injuries, and so on. We were unable to obtain more extensive data, nor did we have access to official medical reports from the soccer clubs. In this regard, we are fully aware that it would be very interesting to compare this information with the data obtained, and this could be a potential future research option. For this reason, we believe that all your proposals are very interesting, but they were not the primary objective of our study, which, as I mentioned at the beginning of this text, is a descriptive observational study.
However, in conclusion, considering the information provided in your comments, we will include these details in the section on limitations and future lines of research.
- Comment1: In the discussion section, can you deepen the goalkeeper HSI? How did this injury happen? Could be a new study perspective?
Response 1: Good observation. However, in the discussion section, we've tried to delve deeper into the mechanisms most common in soccer players, emphasizing their relevance. In the goalkeeper's case, it's not a mechanism exclusive to him; he simply injured himself during a goal kick using a ST-OC mechanism, having to quickly stop a hip flexion with a knee extension, causing a severe strain on the hamstrings.
- Comment 2: Results find that in the middle of the first and the second half, there is a slight increase in HSI. Did it coincide with the part of the game in which players repeat more high-intensity running and, therefore, could be more fatigued? Moreover, results find that the second period with more HSI is at the beginning of the second half. This could be because, on the contrary, players had a lower muscle temperature (if they did not have a warm-up practice during the recovery time) and, therefore, their muscles are more susceptible to injuries. Again, the fewer episodes at the end of the second half could be because there are fewer overall high-intensity actions by the players? Can you expand on these concepts in the discussion?
Response 2: We find this comment very interesting, and thank you for sharing it. However, our study has limitations in this regard, as we didn't have the players' GPS/GNSS data, making it impossible to conduct the comparison you're presenting. Nevertheless, it's an idea worth considering and could be applied in future studies.
- Comment 3: Regarding HSI and curvilinear sprint, why was it not reported whether the injury occurred in the inner or outer leg? Could it be an additional and useful information for future research looking at which leg is more susceptible to HSI in curvilinear sprint?
Response 3: Regarding this comment, the authors fully agree that it would be important to know this information, as it has already been investigated in the literature that the inside and outside legs of a curved sprint perform different mechanical actions. However, again, we did not have access to the football clubs' medical reports, and the TransferMarkt platform often did not indicate which leg was injured, only showing the type of injury. For this reason, we decided not to include this information, although it could clearly be something to consider in future research.
- Comment 4: What was the age of the injured players? Could it be an useful information to stratify your results? Do older players suffer prevalently from a certain type of HSI than younger players?
Response 4: Regarding the average age of the players, it is presented in the results section, in the "Subjects" subsection, indicating that it was 28 ± 3.91 years. However, your second observation may be something worth considering, so we will consider whether a future study can carry out some categorization regarding age and injury mechanism. Even so, as we have previously indicated, this was not the objective of this study, as it was intended to focus on injury mechanisms.
- Comment 5: Additionally, some of these HSI were they a re-injury? Again, could it be another variable to stratify the results?
Response 5: Another excellent comment, and one that would undoubtedly be a variable to consider. However, as I mentioned earlier, we didn't have access to the clubs' medical reports (epidemiological data), so we didn't have that information. Clearly, this information is one of the main limitations of our study, so it should be noted in the limitations section, as it is valuable information that could be included in future research.
- Comment 6: Finally, considering the easy-to-access data (video recording in open access) and no requirement of an ethics form, why did you only consider two seasons and not a longer period (such as 5 years) as in one of the studies cited as reference?
Response 6: Thank you very much for this comment. Here we have to differentiate between access to match videos and access to players' personal data. The second case was freely accessible, as it was available on the Transfermarkt website, a site accessible to everyone. However, the videos were exclusive to LaLiga's Mediacoach platform, and we only had access to the two seasons you mentioned, which is why we couldn't include more seasons in the study.
Reviewer 2 Report
Comments and Suggestions for Authors
Dear authors,
First of all, allow me to congratulate you on the work carried out and the methodology employed. With the aim of improving the methodological quality of the study, its relevance and importance, and to promote its applicability, I encourage the authors to implement new inferential statistical analyses to improve the manuscript. Similarly, I suggest a series of modifications to improve the content and interpretation of the data provided in the manuscript.
Regards.
Abstract
- Authors should indicate the study design.
- It is stated that an ad hoc tool was used, but no further details are provided. It would be useful to include this detail in the abstract.
Introduction
- It would be interesting for the authors to point out the main risk factors (beyond the study objective).
- The manuscript focuses on male soccer players. It would be interesting to see a description of why male players were chosen (justification), whether previous differences have been described based on gender and categories (professional/non-professional).
- Considering the evidence available to date, with the latest studies correctly described by the authors (references 28, 31, etc.), it would be relevant for the authors to state the study hypothesis before detailing the study objective.
Methods
- Were there no exclusion criteria for determining which athletes were included in the study?
- One of the requirements for research today is that the research project must be registered in a database beforehand. This aspect does not appear in the Methods section. Did the authors register the study prospectively? If so, they should indicate this, and if not, they should do so retrospectively.
- Was the evaluation of the two researchers carried out jointly or independently, with the results of the analysis then being pooled? This aspect is key to the methodological quality of the study and the authors should point this out in the manuscript.
- The authors provide a descriptive analysis of the injuries during the study seasons. However, I believe that more conclusive data could be obtained from the work carried out. I encourage the authors to carry out a more exhaustive analysis of the data, determining, for example, the risk based on variables such as age, playing time, position on the field, etc., calculating inferential statistics such as relative risk, incidence, and prevalence, and even an analysis to obtain a predictive model of injury.
Results
- As I pointed out in the previous section, I believe that the results should be implemented with inferential analysis to get the most out of the research.
Discussion
- This section is very informative, easy to read, and implement. However, as the authors indicate in the limitations section, if inferential analysis were implemented, the content of this section would be much more enriching.
- I encourage the authors to be more self-critical in describing the limitations of the study, as this aspect is frankly very poor.
Conclusions
- Greater syntactic skill in the drafting of the conclusions would be desirable. Some statements are overly categorical, given the limitations acknowledged by the authors themselves and the fact that they are based exclusively on descriptive data.
Author Response
First of all, thank you very much for your comments. They provide very interesting additional information for our study, and we will take them all into consideration. We have observed that the vast majority of them refer to the methodological section of the study, so we believe it is appropriate to write this brief introduction to clarify most of the questions raised.
When we began working on this study, the main objective was to describe in more detail the injury mechanisms of the hamstring muscles in soccer players, as until then there was little descriptive information about them. To this end, we contacted LaLiga so they could provide us with videos of some sports seasons through the Mediacoach platform (which includes the tactical and television cameras). They agreed to work with us, but we had a time limit (6 months) to complete the entire video filtering process, include all injuries that were solely related to the hamstring muscles, generate the ad hoc observational tool, analyze the observability of injuries, and so on. We were unable to obtain more extensive data, nor did we have access to official medical reports from the soccer clubs. In this regard, we are fully aware that it would be very interesting to compare this information with the data obtained, and this could be a potential future research option. For this reason, we believe that all your proposals are very interesting, but they were not the primary objective of our study, which, as I mentioned at the beginning of this text, is a descriptive observational study.
However, in conclusion, considering the information provided in your comments, we will include these details in the section on limitations and future lines of research.
We hope to make the proposed changes as soon as possible and resubmit the manuscript to the journal.
Comments 1: Authors should indicate the study design. It is stated that an ad hoc tool was used, but no further details are provided. It would be useful to include this detail in the abstract.
Response 1: Good observation. We fully agree that we should indicate the study design, an important detail we haven't mentioned. Thank you very much for the comment. However, regarding the ad hoc observation tool, the entire description is described in the supplementary data, as it was a lot of information to include in the study. Regarding the abstract, we didn't want to include more information due to the word limit.
Comments 2: Were there no exclusion criteria for determining which athletes were included in the study?
Response 2: There were no exclusion criteria for players. In other words, the objective was to include as many soccer players with hamstring injuries as possible to obtain a larger sample size. Again, the objective of this study was to describe injury mechanisms and contextualize them during official matches, without considering other types of external data.
Comments 3: One of the requirements for research today is that the research project must be registered in a database beforehand. This aspect does not appear in the Methods section. Did the authors register the study prospectively? If so, they should indicate this, and if not, they should do so retrospectively.
Response 3: Thank you very much for the comment. Again, this is an important piece of information we need to include in the methods section. Thanks for sharing it.
Comments 4: Was the evaluation of the two researchers carried out jointly or independently, with the results of the analysis then being pooled? This aspect is key to the methodological quality of the study and the authors should point this out in the manuscript.
Response 4: The researchers' assessment was conducted independently. Initially, the categories and criteria to be observed were defined based on the literature available at the time. The two observers then independently analyzed 25 selected sequences, establishing inter-observer agreement in any uncertain situations. All injuries for which there was no consensus or were unobservable were excluded. Regarding the analysis, Cohen's kappa (k) is described to assess the agreement between the two evaluators in the "video analysis" section.
Comments 5: The authors provide a descriptive analysis of the injuries during the study seasons. However, I believe that more conclusive data could be obtained from the work carried out. I encourage the authors to carry out a more exhaustive analysis of the data, determining, for example, the risk based on variables such as age, playing time, position on the field, etc., calculating inferential statistics such as relative risk, incidence, and prevalence, and even an analysis to obtain a predictive model of injury.
Response 5: As previously mentioned in the text, this is a descriptive, observational study that describes the injury mechanisms of hamstring injuries in male soccer players. We are aware that this is not an exhaustive study of injuries, but rather simply an observation of injury mechanisms in general. Even so, we did conduct an exploratory inferential analysis, but due to the distribution of the data (by position, time of play, type of mechanism, etc.), no statistically significant results were found. Furthermore, as also mentioned above, we did not have access to data other than the videos, and access to these was time-limited, so we cannot currently address many of the issues raised in these reviews.
Comments 6: Greater syntactic skill in the drafting of the conclusions would be desirable. Some statements are overly categorical, given the limitations acknowledged by the authors themselves and the fact that they are based exclusively on descriptive data.
Response 6: Thanks for the comment. We'll try to change the writing of the conclusions section.